# A neural circuit linking learning and sleep in *Drosophila* long-term memory

Zhengchang Lei[1], Kristin Henderson [1] & Krystyna Keleman [1✉]

Animals retain some but not all experiences in long-term memory (LTM). Sleep supports LTM retention across animal species. It is well established that learning experiences enhance post-learning sleep. However, the underlying mechanisms of how learning mediates sleep for memory retention are not clear. *Drosophila* males display increased amounts of sleep after courtship learning. Courtship learning depends on Mushroom Body (MB) neurons, and post-learning sleep is mediated by the sleep-promoting ventral Fan-Shaped Body neurons (vFBs). We show that post-learning sleep is regulated by two opposing output neurons (MBONs) from the MB, which encode a measure of learning. Excitatory MBONs-γ2α′1 becomes increasingly active upon increasing time of learning, whereas inhibitory MBONs-β′2mp is activated only by a short learning experience. These MB outputs are integrated by SFS neurons, which excite vFBs to promote sleep after prolonged but not short training. This circuit may ensure that only longer or more intense learning experiences induce sleep and are thereby consolidated into LTM.

---

[1] Janelia Research Campus, HHMI, 19700 Helix Dr, Ashburn, VA 20147, USA. ✉email: kelemank@janelia.hhmi.org

Post-learning sleep supports LTM retention across animal species[1–5]. It mediates consolidation of spatial and courtship LTM in rodents and fruit flies, respectively, by the reactivation of the neural pathways previously engaged during memory acquisition[6–8]. One universal and well-studied[9–11] feature of sleep is homeostatic control, which balances sleep need after sleep loss[12,13]. Sleep need is also mediated by learning[4,8,14,15]. However, it is not clear how learning induces sleep for memory consolidation.

*Drosophila* males that engage in prolonged and futile 6-h courtship of unreceptive mated females display enhanced post-learning sleep in a time period between 1–3 h (TP1-3) after the end of training[4,8]. This post-learning sleep is induced by learning, not by fatigue following long and vigorous courtship, as males mutant for the dopamine receptor DopR1, which can't learn but court-mated females during training as actively as the wild-type males, do not exhibit increased sleep[8]. Post-learning sleep is essential to consolidate such a courtship experience into LTM[4,8], which is expressed as a courtship suppression towards mated females[16–18] and persists for at least 24 h.

Courtship learning depends on the Mushroom Body (MB) neurons[19–23], and post-learning sleep is mediated by the sleep-promoting ventral Fan-Shaped Body neurons (vFBs)[8]. vFBs reactivate the dopaminergic neurons that were active during memory acquisition in the TP1-3 after learning for the memory to persist[8]. In this study, we aimed to understand how learning regulated by the MB is conveyed to vFBs to induce post-learning sleep for consolidation of LTM.

Here, we identify a neural circuit which connects the MB with the vFB neurons. Integrated activity of the two antagonistic MB output neurons in SFS neurons may ensure that only long learning experiences induce post-learning sleep and are thus consolidated to LTM.

## Results

**Long but not short learning experience activates vFBs.** Post-learning sleep is generated after courtship experience that leads to long- but not short-lasting memory[4,8]. We hypothesized that this selectivity is reflected in the activity of vFBs, such that they would be selectively activated in TP1-3 after a learning experience that induces LTM but not short-term memory (STM). To test this hypothesis, we expressed a luminescence-based transcriptional reporter of neuronal activity, Lola, in vFBs (Supplementary Fig. 1a), which allowed us to measure their activity in freely behaving males[24,25]. Individual males were subjected to varying periods of training, for either 1, 2, 4 or 6 h, with mated females. All trainings were performed in parallel and aligned at the end of 6-h training (from −6 to 0 h). Afterward, the luminescence signal was measured for several h in 15-min intervals. Control naïve males did not undergo training. As predicted, the activity of vFBs was significantly higher in TP1-3, with the maximum activity in TP1-2, after a 6-h training, which is typically used to induce LTM. Somewhat surprisingly, vFBs activity was also significantly higher after training for 4 h in comparison to the control naïve males and males that were trained for 1 or 2 h (Fig. 1a, b). This effect was specific to vFBs, as training with mated females for 6 h did not lead to increased activity of the dorsal FB neurons (dFBs), which are involved in sleep homeostasis[5,9,10,26] (Supplementary Fig. 2a). Consistent with the results in freely behaving males, activity of vFBs, as measured by two-photon imaging of $Ca^{++}$ levels in in vivo brain preparations, was significantly higher throughout the entire TP1-3 in males subjected to 6- but not 1-h training with mated females relative to control males (Supplementary Fig. 2b–f). This broader peak of vFBs activity, in comparison to the luminescence assay, likely reflects a higher

sensitivity of the GCaMP7b-based assay. All together, these results suggest that vFBs are selectively activated by a learning experience that is sufficient to induce LTM.

Accordingly, post-learning activity of vFBs is selectively required for LTM but not STM. We blocked the activity of vFBs with the temperature sensitive blocker of neurotransmission Shibire (Shi$^{ts}$), which is active at 30 °C but not 20 °C, after 1- or 6-h training to induce STM or LTM, respectively. Due to rather poor temporal resolution of Shi$^{ts}$ and to ensure we silenced the neurons in the entire time window of interest, we extended the incubation time at 30 °C for 30 min at each end. Males with vFBs silenced at 30 °C had specifically impaired LTM relative to males that remained at 20 °C and genetic controls in both temperatures, which all formed normal STM and LTM (Supplementary Fig. 2g, h).

If vFBs are indeed activated only by learning experiences that induce LTM, then post-learning sleep and robust LTM should be generated not only by 6-h but by 4-h training as well, but not shorter training. We exposed naïve males to mated females for 1, 2, 4 or 6 h in single pair assays and measured the amount of sleep for the rest of the day. Control naïve males did not experience mated females during this time. To monitor sleep, we used a video tracking system (SleepTracker) which monitors the movements of males with high spatial and temporal resolution. Consistent with the results in Fig. 1a, b, we observed a robust sleep increase in the TP1-3 after 4 and 6 h of training relative to naïve males or those trained for only 1 or 2 h (Fig. 1c, d). These males also sleep more deeply, as evidenced by the higher probability of them falling asleep and the lower probability of waking up during this time, compared to males that were trained only for 1 or 2 h[27] (Supplementary Fig. 2i, j). To assess LTM, we measured the extent of courtship suppression of the identically trained males towards single freshly mated females in a 10-min test, 24 h after training. Accordingly, males that were trained for 4 or 6 h formed robust LTM quantified as a courtship suppression index, SI [%], (Fig. 1e).

Since vFBs display increased activity in TP1-3 after a prolonged learning experience, we expected their activity during this time to be critical to generate post-learning sleep. We thus silenced vFBs (Supplementary Fig. 1a, b) with Shi$^{ts}$. Males were trained for 6 h with mated females, and vFBs were silenced in TP1-3 after training. Males incubated at 30 °C did not display a sleep enhancement, while males that remained at 20 °C, as well as the genetic control groups tested at 20 °C and 30 °C, showed the expected sleep increase (Fig. 1f). These results demonstrate that vFBs are essential for post-learning sleep and are selectively activated by a learning experience that is sufficient to induce LTM.

Post-learning sleep is induced specifically by prolonged learning[8], and the increasing duration of a learning experience correlates with the total amount of time males spend courting mated females during training (Supplementary Fig. 2k). If vFBs are to sense the amount or intensity of a learning experience, they should respond to upstream neurons that convey a measure of the learning experience needed for LTM formation. Since, courtship learning critically depends on MB neurons (MBs)[20], we hypothesized that specific MBs encode and convey the amount of learning onto vFBs. We focused on MB output neurons (MBONs), believed to mediate the behavioral outputs induced by different forms of learning[28]. We hypothesized that specific MBONs involved in post-learning sleep and memory consolidation would fulfill four criteria: they should (1) promote sleep, (2) be necessary for LTM consolidation and the post-learning sleep enhancement, (3) provide excitatory input to vFBs, and (4) be activated by a learning experience.

**Activity of MBONs-γ2α'1 reflects the length of learning.** A small set of MBONs was previously shown to promote sleep upon

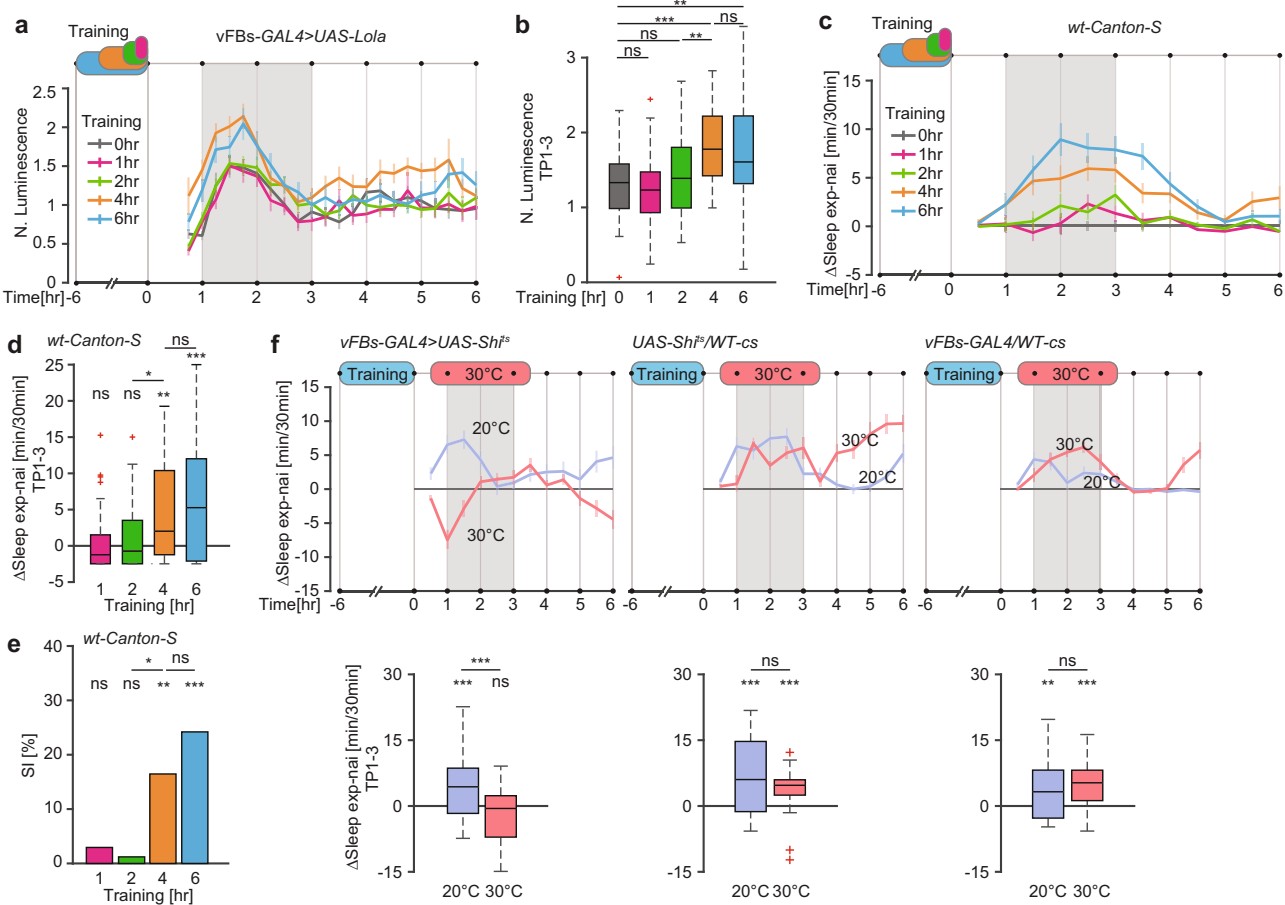

**Fig. 1 vFBs are activated after long but not short courtship experience. a** Mean normalized luminescence traces (±SEM) in vFBs. **b** Mean normalized luminescence in the 1–3 h time period after training (TP1-3) in **a**. $n_{0\,h} = 33$, $n_{1\,h} = 29$, $n_{2\,h} = 37$, $n_{4\,h} = 35$, $n_{6\,h} = 35$, $P_{1\,h} = 0.7824$, $P_{2\,h} = 0.2996$, $P_{4\,h} = 2.9e-5$, $P_{6\,h} = 0.0065$ for $H_0$ N. Lum.$_{exp}$ = N. Lum.$_{nai}$, $P = 0.0014$ for $H_0$ N. Lum.$_{2\,h}$ = N. Lum.$_{4hr}$, $P = 0.6311$, for $H_0$ N. Lum.$_{4\,h}$ = N. Lum.$_{6\,h}$, two-sided Student $T$ test. **c** Mean sleep traces of $\Delta$Sleep (exp-nai) (±SEM) after training. **d** Mean $\Delta$Sleep (exp-nai) in TP1-3 in **c**. $n_{0\,h} = 45$, $n_{1\,h} = 42$, $n_{2\,h} = 41$, $n_{4\,h} = 44$, $n_{6\,h} = 48$, $P_{1\,h} = 0.5480$, $P_{2\,h} = 0.6338$, $P_{4\,h} = 0.0022$, $P_{6\,h} = 1.1e-4$ for $H_0$ $\Delta$Sleep = 0, two-sided Wilcoxon Signed Rank test, and $P = 0.0251$ for $H_0$ $\Delta$Sleep$_{2\,h}$ = $\Delta$Sleep$_{4\,h}$, $P = 0.4809$ for $H_0$ $\Delta$Sleep$_{4\,h}$ = $\Delta$Sleep$_{6\,h}$, two-sided Wilcoxon Rank Sum test. **e** LTM, shown as Suppression Index, SI [%] of males trained as indicated in **a**. $n_{0\,h} = 58$, $n_{1\,h} = 60$, $n_{2\,h} = 59$, $n_{4\,h} = 63$, $n_{6\,h} = 60$, $P_{1\,h} = 0.2942$, $P_{2\,h} = 0.3277$, $P_{4\,h} = 0.0022$, $P_{6\,h} = 1.2e-4$ for $H_0$ SI = 0, and $P = 0.0175$ for $H_0$ SI$_{2\,h}$ = SI$_{4\,h}$, two-sided Permutation test. **f** (top) Mean sleep traces of $\Delta$Sleep (exp-nai) (±SEM) upon vFBs silencing. (bottom) Mean $\Delta$Sleep (exp-nai) in TP1-3 of the data at the top. (left) $n_{20\,°C} = 57$, $n_{30\,°C} = 59$, $P_{20\,°C} = 1.4e-4$, $P_{30\,°C} = 0.0581$ value is for $H_0$ $\Delta$Sleep = 0, and $P = 2.6e-5$ for $H_0$ $\Delta$Sleep$_{20\,°C}$ = $\Delta$Sleep$_{30\,°C}$, (middle) $n_{20\,°C} = 63$, $n_{30\,°C} = 46$, $P_{20\,°C} = 1.6e-6$, $P_{30\,°C} = 1.4e-6$ for $H_0$ $\Delta$Sleep = 0, and $P = 0.1399$ for $H_0$ $\Delta$Sleep$_{20\,°C}$ = $\Delta$Sleep$_{30\,°C}$, (right) $n_{20\,°C} = 59$, $n_{30\,°C} = 65$, $P_{20\,°C} = 0.0019$, $P_{30\,°C} = 1.3e-9$ for $H_0$ $\Delta$Sleep = 0, and $P = 0.0612$ for $H_0$ $\Delta$Sleep$_{20\,°C}$ = $\Delta$Sleep$_{30\,°C}$. Two-sided Wilcoxon Signed Rank test for single groups and two-sided Wilcoxon Rank Sum test across groups. Full genotypes and data analysis details can be found in Supplementary Tables 1–3. Source data are provided as a Source Data file. ns $P > 0.05$, *$P < 0.05$, **$P < 0.01$, ***$P < 0.001$. $n$ represents independent fly samples with assays repeated at least 3 times (**b**, **d**, **e** and **f**). Box plots represent median and IQR and whiskers extend to lower and upper adjacent values (**b**, **d** and **f**).

extended thermogenetic activation[28]. Since post-learning sleep is limited to a short time window after learning, we used a shorter photoactivation protocol to identify MBONs able to induce sleep acutely. We individually activated all 21 MBON classes[22] with csChrimson for 1 h. As positive controls we included vFBs and dFBs, which mediate post-learning and homeostatic sleep, respectively[5,8–10,26]. We identified two sleep-promoting MBONs classes[20] that acutely induce sleep: both MBONs-γ2α'1 and MBONs-calyx neurons enhanced sleep upon light stimulus in comparison to animals that were not illuminated (Fig. 2a). Given that analysis of the Focused Ion Beam Scanning Electron Microscopy (FIB-SEM) volume of the *Drosophila* brain[29,30] revealed that MBONs-γ2α'1, but not MBONs-calyx, might be functionally connected to vFBs via one class of interneurons (Fig. 2b), we focused on MBONs-γ2α'1. To test if MBONs-γ2α'1 are necessary for LTM and enhanced sleep after

learning, we expressed Shi[ts] specifically in MBONs-γ2α'1 (Supplementary Fig. 1c). We trained males for 6 h with mated females and silenced MBONs-γ2α'1 after training. Males with MBONs-γ2α'1 silenced at 30 °C did not display LTM or increased sleep when compared to naïve males (Fig. 2c, d). In contrast, the experimental males that remained at 20 °C and the genetic control males at either temperature, all formed normal LTM and slept more than naïve males after the learning experience.

Activation of the cholinergic and thus presumably excitatory MBONs-γ2α'1[28] with Chrimson88 elicited excitatory responses in vFBs but not dFBs in two-photon calcium imaging experiments using the calcium indicator GCaMP6s. There was no response in vFBs upon activation of the sleep-promoting MBONs-calyx, which based on the data extracted from the FIB-SEM volume of the Drosophila brain[29,30] have no immediate connection to vFBs (Fig. 2e, Supplementary Fig. 1d, f). In support of these results,

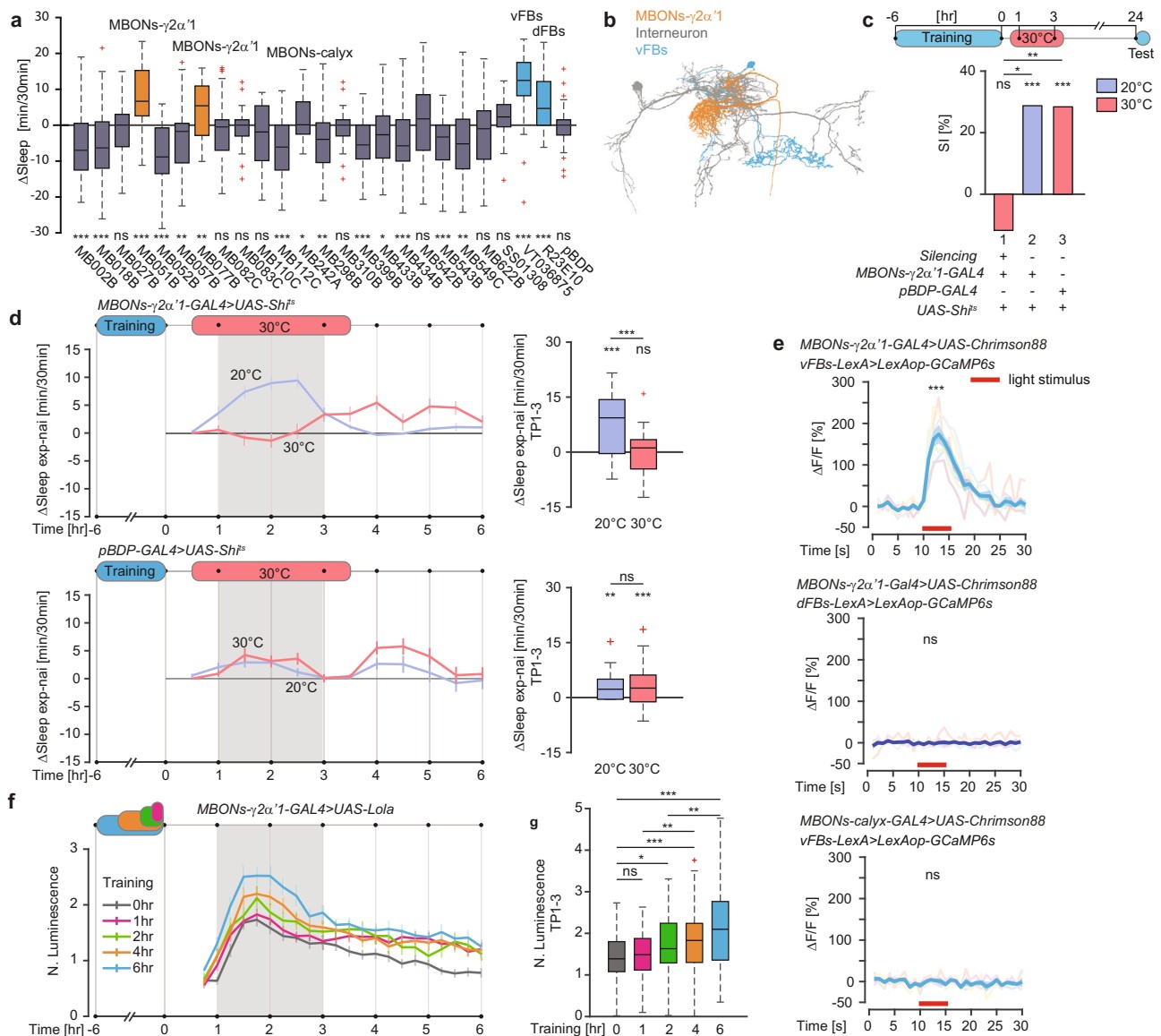

**Fig. 2 MBONs-γ2α'1 activity reflects the length of learning. a** Mean sleep change upon optogenetic activation of MBONs. $P$ value is for $H_0$ ΔSleep(light-nolight) = 0, two-sided Wilcoxon Signed Rank test, $n = 36–54$ per group (Supplementary Table 3). **b** Neural skeletons derived from the FIB-SEM *Drosophila* brain volume. **c** LTM, shown as Suppression Index, SI [%], upon MBONs-γ2α'1 silencing. $n_1 = 53$ and 51, $n_2 = 53$ and 54, $n_3 = 44$ and 44 for naïve and trained group respectively, $P_1 = 0.8602$, $P_2 = 0.0003$, $P_3 = 0.0001$ for $H_0$ SI = 0, and $P_{12} = 0.0105$, $P_{13} = 0.0013$ for $H_0$ SI$_{exp}$ = SI$_{ctrl}$, two-sided Permutation test. **d** (left) Mean sleep traces of ΔSleep (exp-nai) (±SEM) upon MBONs-γ2α'1 silencing. (right) Mean ΔSleep (exp-nai) in the 1–3 h time period after training (TP1-3) on the left. (top) $n_{20\,°C} = 70$, $n_{30\,°C} = 46$, $P_{20\,°C} = 5.3e–8$, $P_{30\,°C} = 0.9608$ for $H_0$ ΔSleep = 0, and $P = 3.9e–6$ for $H_0$ ΔSleep$_{20\,°C}$ = ΔSleep$_{30\,°C}$. (bottom) $n_{20\,°C} = 68$, $n_{30\,°C} = 69$, $P_{20\,°C} = 0.0065$, $P_{30\,°C} = 0.0002$ for $H_0$ ΔSleep = 0, and $P = 0.2388$ for $H_0$ ΔSleep$_{20\,°C}$ = ΔSleep$_{30\,°C}$. Two-sided Wilcoxon Signed Rank test for single groups and two-sided Wilcoxon Rank Sum test across groups. **e** Mean GCaMP6s response (ΔF/F) traces (±SEM) in vFBs (top), dFBs (middle) upon optogenetic activation of MBONs-γ2α'1 (red bar) or in vFBs (bottom) upon MBONs-calyx activation (red bar). $n_{top} = 7$, $n_{middle} = 4$, $n_{bottom} = 7$, $P_{top} = 1.0e–5$, $P_{middle} = 0.6254$, $P_{botton} = 0.4812$, for $H_0$ ΔF/F = 0, two-sided Student $T$ test. **f** Normalized mean luminescence traces (±SEM). **g** Mean normalized luminescence in TP1-3 in **f**. $n_{0\,h} = 73$, $n_{1\,h} = 84$, $n_{2\,h} = 50$, $n_{4\,h} = 61$, $n_{6\,h} = 67$, $P_{1\,h} = 0.1476$, $P_{2\,h} = 0.0214$, $P_{4\,h} = 7.7e–4$, $P_{6\,h} = 8.0e–8$ for $H_0$ N. Lum.$_{exp}$ = N. Lum.$_{nai}$, $P = 0.0068$ for $H_0$ N. Lum.$_{1hr}$ = N. Lum.$_{4\,h}$, and $P = 0.0050$ for $H_0$ N. Lum.$_{2\,h}$ = N. Lum.$_{6\,h}$, two-sided Student $T$ test. Full genotypes and data analysis details in Supplementary Tables 1–3. Source data are provided as a Source Data file. ns $P > 0.05$, *$P < 0.05$, **$P < 0.01$, ***$P < 0.001$. $n$ represents independent fly samples with assays repeated at least 3 times (**a**, **c**, **d** and **g**). Box plots represent median and IQR, whiskers extend to lower and upper adjacent values and red crosses for outliers (**a**, **d** and **g**).

silencing MBONs-γ2α'1 with Shi$^{ts}$ while monitoring activity of vFBs with the luminescence-based Lola reporter in freely behaving males blocked the vFBs activity increase in TP1-3 after 6-h training (Supplementary Figs. 1j and 3). In summary, these data suggest that MBONs-γ2α'1 mediate post-learning sleep required for LTM by providing an excitatory input to vFBs. These results, together with our data showing that the vFB and dFB neurons are not functionally connected (Supplementary Fig. 4a,

b) and dFBs have no role in courtship LTM[8], suggest that learning-induced sleep and homeostatic sleep regulation operate independently of each other.

To test whether MBONs-γ2α'1 activity represents a measure of prolonged learning experience, we trained males for 1, 2, 4 or 6 h with mated females and monitored MBONs-γ2α'1 activity with the luminescence-based reporter, Lola. We observed that MBONs-γ2α'1 are indeed activated upon learning. Interestingly,

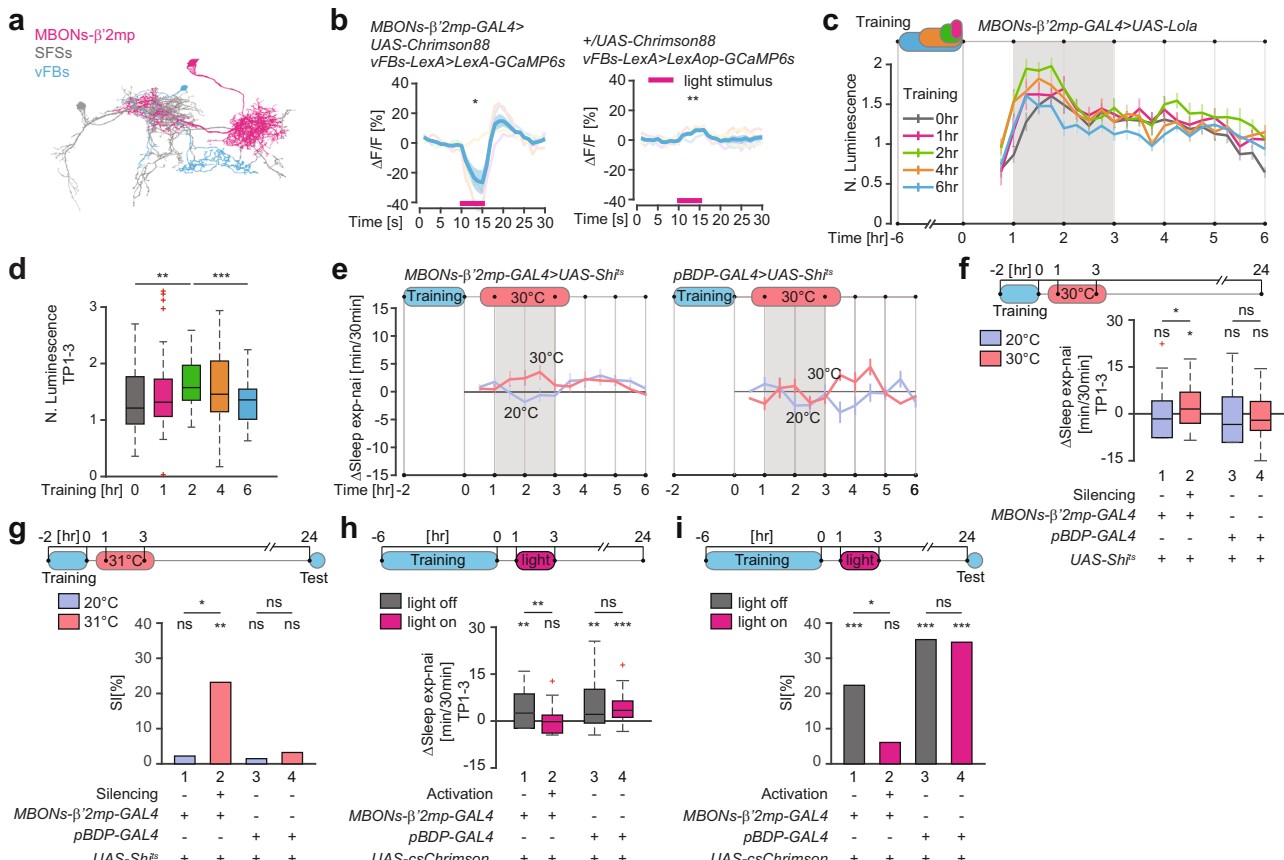

**Fig. 3 MBONs-β'2mp activity peaks after short training. a** Neural skeletons derived from the FIB-SEM *Drosophila* brain volume. **b** Mean GCaMP6s response (ΔF/F) traces (±SEM) in vFBs upon optogenetic MBONs-β'2mp activation (red bar). $n = 5$ flies per group, $P_{left} = 0.0303$, $P_{right} = 0.0071$ for $H_0$ ΔF/F = 0, two-sided Student *T* test. **c**. Normalized mean luminescence traces (±SEM). **d**. Mean normalized luminescence in the 1–3 h time period (TP1-3) after training in **c**. $n_{0h} = 47$, $n_{1h} = 38$, $n_{2h} = 39$, $n_{4h} = 44$, $n_{6h} = 38$, $P_{1h} = 0.2769$, $P_{2h} = 0.0056$, $P_{4h} = 0.1043$, $P_{6h} = 0.8331$ for $H_0$ N. Lum.$_{exp}$ = N. Lum.$_{nair}$, and $P = 8.6e-4$ for $H_0$ N. Lum.$_{2h}$ = N. Lum.$_{6h}$, two-sided Student *T* test. **e** Mean sleep traces (±SEM) of ΔSleep (exp-nai) upon MBONs-β'2mp silencing. **f** Mean ΔSleep (exp-nai) in TP1-3 in **e**. $n_1 = 54$, $n_2 = 49$, $n_3 = 39$, $n_4 = 42$, $P_1 = 0.6073$, $P_2 = 0.0207$, $P_3 = 0.3553$, $P_4 = 0.3451$ for $H_0$ ΔSleep = 0, two-sided Wilcoxon Signed Rank test, and $P_{12} = 0.0273$, $P_{34} = 0.3559$ for $H_0$ ΔSleep$_{20°C}$ = ΔSleep$_{30°C}$, two-sided Wilcoxon Rank Sum test. **g** LTM, shown as SI [%] upon MBONs-β'2mp silencing. $n_1 = 53$ and 52, $n_2 = 59$ and 61, $n_3 = 52$ and 57, $n_4 = 72$ and 59 for naïve and trained group respectively, $P_1 = 0.2188$, $P_2 = 0.0019$, $P_3 = 0.4019$, $P_4 = 0.2663$ for $H_0$ SI = 0, and $P_{12} = 0.0376$, $P_{34} = 0.8575$ for $H_0$ SI$_{exp}$ = SI$_{ctrl}$, two-sided Permutation test. **h** Mean ΔSleep (exp-nai) in TP1-3 upon MBONs-β'2mp activation. $n_1 = 40$, $n_2 = 40$, $n_3 = 40$, $n_4 = 38$, $P_1 = 0.0025$, $P_2 = 0.6374$, $P_3 = 0.0014$, $P_4 = 1.2e-5$ for $H_0$ ΔSleep = 0, two-sided Wilcoxon Signed Rank test, and $P_{12} = 0.0023$, $P_{34} = 0.8768$ for $H_0$ ΔSleep$_{noact}$ = ΔSleep$_{act}$, two-sided Wilcoxon Rank Sum test. **i** LTM, shown as SI [%], upon MBONs-β'2mp activation. $n_1 = 56$ and 57, $n_2 = 57$ and 56, $n_3 = 30$ and 40, $n_4 = 33$ and 37 for naïve and trained group respectively, $P_1 = 1.0e-5$, $P_2 = 0.0517$, $P_3 = 4.8e-4$, $P_4 = 1.0e-5$ for $H_0$ SI = 0, and $P_{12} = 0.0403$, $P_{34} = 0.9542$ for $H_0$ SI$_{exp}$ = SI$_{ctrl}$, two-sided Permutation test. Full genotypes and data analysis details in Supplementary Tables 1–3. Source data are provided as a Source Data file. ns $P > 0.05$, *$P < 0.05$, **$P < 0.01$, ***$P < 0.001$. n represents independent fly samples with assays repeated at least 3 times (**d**, **f**, **g**, **h** and **i**). Box plots represent median and IQR, whiskers extend to lower and upper adjacent values and red crosses for outliers (**d**, **f** and **h**).

their activity increased linearly with the length of learning (Fig. 2f, g). These results suggest that MBONs-γ2α'1 activity reflects the amount of the learning experience in a linear fashion. However, this linear signal must somehow be transformed into the all-or-none stepwise activation of vFBs upon prolonged learning experience.

**Short but not long learning experience activates MBONs-β'2mp.** In our screen to identify sleep-promoting MBONs, we found one potential candidate class of neuron that could contribute to this transformation. Analysis of the FIB-SEM volume revealed that out of several MBONs that suppress sleep acutely upon optogenetic activation with csChrimson, MBONs-β'2mp (MB002B) (Fig. 2a, Supplementary Fig. 1h) appear to be connected to vFBs via the same SFS neurons (SFSs) as MBONs-γ2α'1 (Fig. 3a). Earlier studies determined that MBONs-β'2mp are glutamatergic[28], and in two-photon imaging experiments we

detected robust inhibitory responses in vFBs upon photoactivation of MBONs-β'2mp (Fig. 3b).

To investigate whether and how MBONs-β'2mp contribute to the vFBs selective activation upon prolonged training, we trained naïve males expressing the Lola reporter in MBONs-β'2mp for either 1, 2, 4 or 6 h with mated females and measured the luminescence signal afterward. Activity of MBONs-β'2mp peaked after training for 2 h, with little or no increase after training for either 4 or 6 h (Fig. 3c, d). These results suggest that MBONs-β'2mp inhibit vFBs and thus suppress sleep after a short learning experience (2 h). This would allow for vFBs activation by MBONs-γ2α'1 when that inhibition is released, and thus allow sleep to occur only after longer experiences (4 or 6 h of training).

The only available LexA line, which would allow us to silence MBONs-β'2mp while simultaneously monitoring vFBs activity in freely behaving males, is also expressed in MBONs-γ5β'2a, which are essential for memory acquisition[31] (Supplementary Fig. 1k).

Thus, we lacked the genetic tools needed to test directly whether activity of MBONs-β'2mp inhibits vFBs after training for STM. We hypothesized however that silencing of MBON-β'2mp neurons after 2 h of training, which normally does not induce sleep and results in only STM, should disinhibit vFBs and thereby allow both post-learning sleep and the consolidation of STM to LTM. To test this, we expressed Shi[ts] in MBONs-β'2 mp. Indeed, males in which MBONs-β'2mp were silenced at 30 °C displayed a significant increase in both post-learning sleep (Fig. 3e, f) and LTM (Fig. 3g) in comparison to naïve males, genetic controls, and males that remained at 20 °C. Conversely, optogenetic activation of MBONs-β'2mp and thus inhibiting vFBs after training for 6 h, suppressed sleep (Fig. 3h) and impaired LTM (Fig. 3i). All together, these results imply that MBONs-β'2mp inhibit vFBs after short training and release this inhibition after prolonged training, which leads to selective activation of vFBs by prolonged learning experience. These results suggest that the integrated activity of MBONs-γ2α'1 and MBONs-β'2mp is crucial to generate post-learning sleep for LTM.

**SFSs integrate two antagonistic MB outputs to activate vFBs.** If SFSs integrate both MBONs-γ2α'1 and MBONs-β'2mp inputs (Fig. 4a) to selectively activate vFBs, we reasoned that they should be activated and inhibited by MBONs-γ2α'1 and MBONs-β'2mp, respectively, and they should activate vFBs. Moreover, they should be necessary for post-learning sleep and LTM consolidation, and lastly, they should be selectively activated by prolonged training.

We identified a single GAL4 driver line[32] with expression in, although not limited to, SFSs (Fig. 4b, Supplementary Fig. 1i). Photoactivation of either MBONs-γ2α'1 (Supplementary Fig. 1j) or MBONs-β'2mp (Supplementary Fig. 1k) and two-photon imaging of Ca[++] levels in SFSs revealed excitatory and inhibitory responses, respectively (Fig. 4c, d). To precisely activate SFSs, we used a digital mirror device (DMD)[33] to target illumination activating Chrimson88 specifically in SFSs. Targeted illumination of SFSs elicited excitatory responses in vFBs (Fig. 4e, Supplementary Fig. 4c). These results support our hypothesis that SFSs integrate the two opposing MB outputs for selective activation of vFBs.

To test whether SFSs are essential for post-learning sleep enhancement and LTM, we silenced SFSs with Shi[ts] in TP1-3 after prolonged courtship learning experience. Males with SFSs silenced at 30 °C did not display increased sleep and LTM after prolonged training when compared to males that remained at 20 °C, suggesting that SFSs are essential for post-learning sleep and LTM formation (Fig. 4f, g). If SFSs are indeed to integrate the two opposing inputs to selectively activate vFBs after prolonged training, we predicted that they should be activated in a similar fashion to that of vFBs. To test it, we used two-photon imaging of Ca[++] levels in in vivo brain preparations as a proxy for SFSs activity after training with mated females. Indeed, Ca[++] levels were significantly elevated in SFSs after 4 and 6, but not after 1 or 2 h of training, relative to naïve males (Fig. 4h, i). Thus, the activity pattern of SFSs upon prolonged training resembles that of vFBs. All together, these results suggest that SFS neurons function in the post-learning sleep enhancement for consolidation of LTM by integration of the two opposing inputs from MBs to selectively activate vFBs.

## Discussion

In summary, we have identified a neural circuit that regulates learning-induced sleep for LTM consolidation. This circuit links neurons essential for learning and memory in *Drosophila*, the MB neurons[19–22], with those critical for post-learning sleep, the

vFBs[8]. We propose that only a longer learning experience is sufficient to induce sleep, and thereby be consolidated into LTM. Given that the increasing duration of a learning experience correlates with the total amount of time males spend on futile courtship towards mated females during training, selective activation of vFBs likely depends on the amount or intensity of a learning experience, rather than just its duration. We find that post-learning sleep induction requires integration of two MB outputs, previously implicated in courtship memory[34] in SFSs. Post-learning activity of the excitatory MBONs-γ2α'1 increases linearly with the duration of the prolonged learning experience. In contrast, activity of the inhibitory MBONs-β'2mp peaks after a short experience sufficient to induce STM. As a result, only when the males court-mated females sufficiently long or intensely, the activity of MBONs-γ2α'1 reaches the threshold required to activate SFSs. This in turn leads to activation of vFBs to promote post-learning sleep[8,35,36] and the reactivation of those dopaminergic neurons (DANs) that were involved in memory encoding[8,18,37]. Consequently, biochemical processes essential for LTM consolidation become engaged[37–39] (Fig. 4j).

How might MBONs-γ2α'1 and MBONs-β'2mp measure the learning experience to control post-learning sleep? In homeostatic sleep regulation, the potentiation of R2 neurons reflects a measure of sleep loss that is sensed by dFBs[26], likely in response to the accumulation of byproducts of oxidative stress during sleep loss[40]. In the case of learning-induced sleep, we envision that learning results in lasting changes in the molecular pathways essential for memory formation in the MB. For example, the cAMP pathway along with the dopamine receptor are activated during sleep in a discrete 3-h time window after learning in rodents[36,38] and *Drosophila* males lacking a dopamine receptor, and hence unable to learn, do not display increased post-learning sleep[8]. Thus, the accumulation of changes in the cAMP signaling pathway upon increasing learning experience with mated females might lead to the increasing potentiation of MBONs-γ2α'1 and MBONs-β'2mp after learning. Interestingly, MBONs-γ2α'1 and MBONs-β'2mp display distinct temporal activity patterns upon learning which likely reflects their distinct neuronal properties.

Here, we reveal a circuit mechanism that ensures that only persistent, and thus likely significant, learning experiences generate post-learning sleep to consolidate LTM. Recent findings suggest that dFBs, involved in sleep homeostasis, might mediate a paradoxical type of sleep, in humans also called Rapid Eye Movement (REM) sleep[41]. This in conjunction with our data, provide an opportunity to investigate whether the post-learning sleep, mediated by vFBs, might represent another type of sleep implicated in mammals in memory consolidation[2].

## Methods

***Drosophila* culture conditions**. Flies were reared at 25 °C and 60% humidity in 12 h/12 h light/dark cycle on a standard corn meal food, unless otherwise noted. Virgin male flies were collected after eclosion and aged for 5–6 days, or as indicated otherwise, before assays. Fly strains used in this study are listed in Supplementary Tables 1 and 2.

**Courtship assay**. Courtship conditioning was performed as previously described[17] in 50 well behavioral plates (10 mm in diameter and 5 mm in depth with 150 μl of standard food). Solitary males aged for 5–7 days after eclosion prior to training were placed into wells with (experienced males) or without (naïve males) a single mated female. After training each male was recovered, allowed to rest for 24 h and tested in a new behavioral plate with a fresh mated female. Tests were videotaped for 10 min at 25 hz (Prosilica GT cameras, Allied Vison Technologies) with recording software jVision (Reb. F, Janelia jET, Github). Automated video analysis[42] was used to derive a courtship index (CI) for each male, defined as the percentage of time over a 10 min test period during which the male courts the female. Memory was calculated as a suppression index (SI) that is a relative reduction in the median courtship indices of trained (CI+) versus naïve (CI−) populations: SI = 100*[1 − CI+/CI−]. Mated females for training and test were

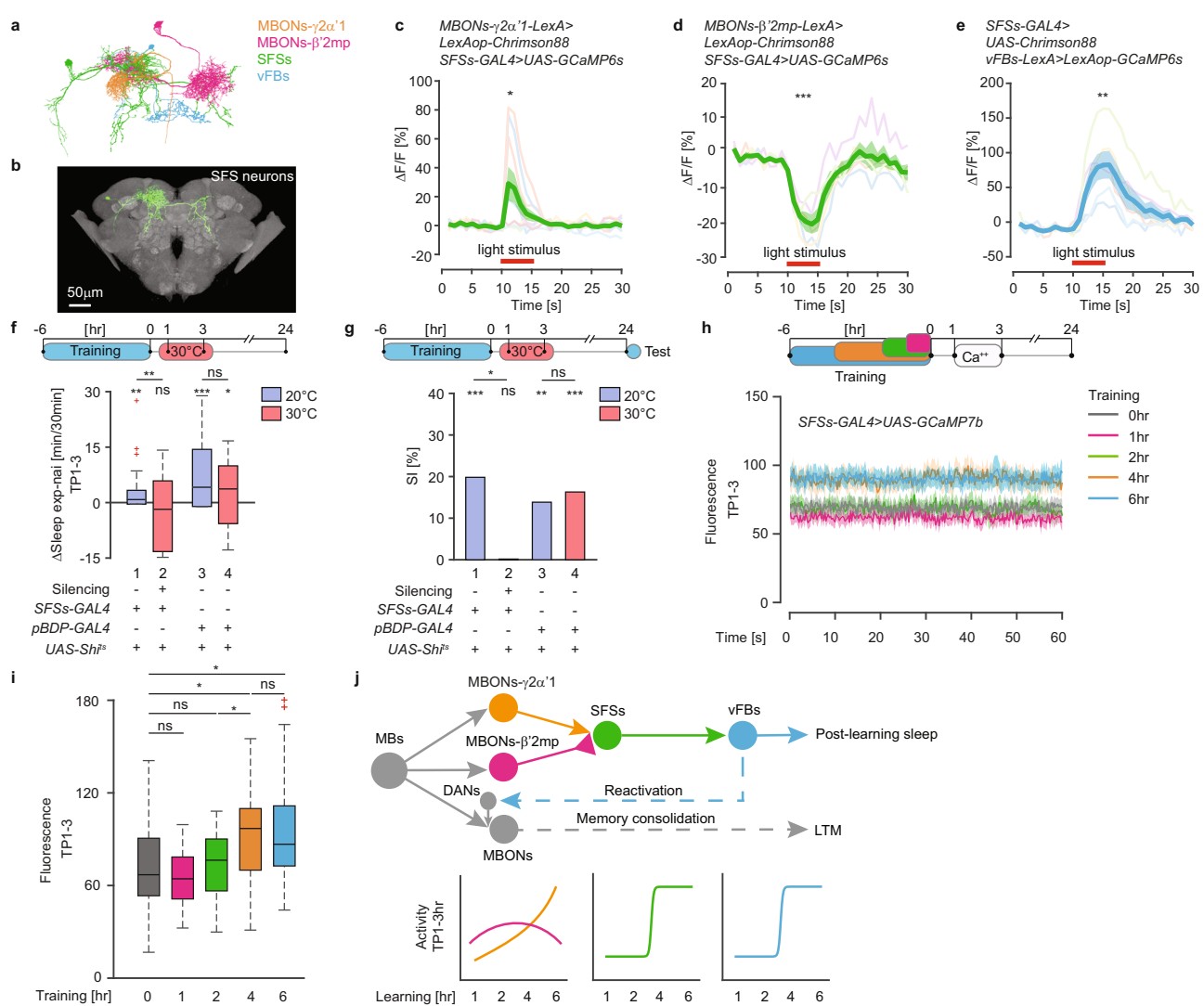

**Fig. 4 SFSs integrate MBONs-γ2α'1 and MBONs-β'2mp inputs onto vFBs. a** Neural skeletons derived from the FIB-SEM *Drosophila* brain volume. **b** Confocal image of SFSs expression pattern extracted from the Multi Color Flip-Out (MCFO) of the *VT033828-GAL4* line. Scale bar = 50 μm. (Details in the Methods). **c, d** Traces of mean GCaMP6s responses (ΔF/F) (±SEM) in SFSs upon optogenetic activation (red bar) of MBONs-γ2α'1 (**c**, $n = 9$, $P = 0.048$), MBONs-β'2mp (**d**, $n = 5$, $P = 0.0002$). *P* value is for $H_0$ ΔF/F = 0, two-sided Student *T* test. **e**. Traces of mean GCaMP6s responses (ΔF/F) (±SEM) in vFBs upon SFSs focal activation (red bar). $n = 6$, $P = 0.0075$ $H_0$ ΔF/F = 0, two-sided Student *T* test. **f**. Mean ΔSleep (exp-nai) in the 1–3 h time period after training (TP1-3) upon SFSs silencing. $n = 39$ flies per group, $P_1 = 0.0046$, $P_2 = 0.1208$, $P_3 = 1.1e{-}4$, $P_4 = 0.0445$ for $H_0$ ΔSleep = 0, two-sided Wilcoxon Signed Rank test, and $P_{12} = 0.0091$, $P_{34} = 0.2711$ for $H_0$ ΔSleep$_{20 °C}$ = ΔSleep$_{30 °C}$, two-sided Wilcoxon Rank Sum test. **g** LTM, shown as SI [%], after SFSs silencing in TP1-3. $n_1 = 52$ and 52, $n_2 = 61$ and 64, $n_3 = 62$ and 51, $n_4 = 65$ and 62 for naïve and trained group respectively, $P_1 = 1.0e{-}5$, $P_2 = 0.4976$, $P_3 = 0.0028$, $P_4 = 2.0e{-}4$ for $H_0$ SI = 0, and $P_{12} = 0.0115$, $P_{34} = 0.7954$ for $H_0$ SI$_{exp}$ = SI$_{ctrl}$; two-sided Permutation test. **h** Mean calcium traces (±SEM) in SFSs, measured for 1 min at multiple time points spanning TP1-3. **i** Mean calcium levels in TP1-3 in **h**. $n_{0 h} = 28$, $n_{1 h} = 17$, $n_{2 h} = 18$, $n_{4 h} = 19$, $n_{6 h} = 26$, $P_{1 h} = 0.3431$, $P_{2 h} = 0.6126$, $P_{4 h} = 0.0312$, $P_{6 h} = 0.0131$ for $H_0$ F$_{nai}$ = F$_{exp}$, $P = 0.0466$ for $H_0$ F$_{2 h}$ = F$_{4 h}$, and $P = 0.8225$ for $H_0$ F$_{4 h}$ = F$_{6 h}$, two-sided Student *T* test. **j** Model of the circuit mechanism to generate post-learning sleep for LTM consolidation. Solid lines indicate monosynaptic connections. Full genotypes and data analysis details in Supplementary Tables 1–3. Source data are provided as a Source Data file. ns $P > 0.05$, *$P < 0.05$, **$P < 0.01$, ***$P < 0.001$. *n* represents independent fly samples with assays repeated at least 3 times (**c, d, e, f, g** and **i**). Box plots represent median and IQR, whiskers extend to lower and upper adjacent values and red crosses for outliers (**f** and **i**).

prepared as follows. Freshly eclosed females and males were housed together for 4 days. Afterward, females were separated from males to be used as mated females.

**Luminescence assay.** The luminescence assay for detecting neuronal activity in freely behaving adult flies was modified from Guo et al.[25]. Solitary males were reared on standard food (cornmeal medium) and aged for 5–7 days after eclosion prior to training. They were trained in 96-well plates (Lumitrac™ 200, Greiner Bio-one, 7 mm in diameter and 11 mm in depth each well) loaded with 35 μl of 40 mM Luciferin (GOLDBIO) in 5% sucrose (Sigma) and 1% agarose (Sigma) or with filter paper soaked with 40 mM Luciferin (GOLDBIO) in a 5% sucrose (Sigma) solution. The plate was covered with plastic film (QIAGEN Cat. No. 195710) and a small notch was cut at each well with a surgical blade to facilitate loading and retrieving

flies from the wells. For luminescence measurements after training, flies were transferred to fresh plates with luciferin containing food or solution and put in a Spark microplate reader (Tecan). In each plate, at least 3 wells were empty, with no flies, for background noise measurement. Luminescence was measured every 15 mins with dual time at each well for 9350 ms over 10 h in 25 °C. The background noise was estimated by the mean luminescence trace of empty wells and subtracted from the raw data of the experienced and control naïve groups. Then the luminescence trace of each fly was normalized (N. Luminescence) by dividing each sampling point by the mean of the average luminescence trace of the naïve group.

**Sleep assay.** To monitor sleep, single males were placed into wells in a customized sleep plate (50 wells, 10 mm in diameter and 5 mm in depth, filed with 150 μl of

standard food). Videos of flies in the sleep plate were recorded at 5 hz (Prosilica GT cameras, Allied Vison Technologies) in gray scale with recording software jVision. Fly movements were traced offline in customized MATLAB program SleepTracker (v1.1, Github)). The locomotion of each fly was measured by the moving distance of a body center over time, which is pooled into 60 s bins with a threshold of 3.6 mm within 60 s. Sleep was defined as the time with no locomotion for 5 mins or more[43,44]. Sleep amount was pooled in 30 min bins. The sleep change (ΔSleep) of the experimental group was calculated by subtracting the mean sleep of the control group from sleep amount of the experimental group at each measurement point for each individual fly.

The probability of falling asleep (P doze) and waking up (P wake) was calculated based on chances of flies transitioning from active to sleep and from sleep to active state respectively[27]. For each 30 min time window, locomotion data was pooled into 30 1 min bins. For each bin, a fly was either in active or non-active status if its locomotion was greater than zero or not. Any bins with 5 or more consecutive minutes of no locomotion was recognized as sleep state, or otherwise wake state. The sleep/wake status of the first and last few bins on the time window edge were judged in the context before and after the time window when there was data available. P doze is the ratio of transitions from active to non-active status, and P wake is the ratio of transitions from non-active to active status. ΔP doze or ΔP wake of the experimental group was calculated by subtracting the mean ΔP doze or ΔP wake trace of control group from each fly's ΔP doze or ΔP wake trace in the experimental group.

To acutely induce sleep with csChrimson, individual males were reared at 25 °C on retinal supplemented (0.1 mM) cornmeal medium in darkness for 4–5 days after eclosion. For sleep induction single males were placed into a 10 mm diameter well behavioral plate in a temperature and illumination-controlled photoactivation chamber and videotaped (Prosilica GT cameras, Allied Vison Technologies) with jVision. Dim white light (0.02 μW/mm²) was illuminated as a background to minimize the startle effect by activation light. For activation, the behavior plate was illuminated by 617 nm LEDs (Red-Orange LUXEON Rebel LED—122 lm; Luxeon Star LEDs) with a 3 mm thick diffuser between the LED and flies. The LED was driven by a customized linear current controller controlled by a customized software RGBctrl (v3.1, Github). Flies were videotaped for 2 h, one group of flies was illuminated in the second hour with ~30 μW/mm² flickering red light at 20 hz while another group was never illuminated with activation light. The amount of sleep was scored in SleepTracker offline. The (ΔSleep) upon red light illumination was calculated by subtracting the mean of the sleep amount in the second hour of non-illuminated flies from the sleep amount of each illuminated fly in the second hour.

**Neuronal silencing with Shibire[ts]**. For neuronal silencing temperature sensitive UAS- Shi[ts] was expressed in target neurons[45]. Experimental and genetic control single housed males were reared on standard food and aged for 7–10 days[46] in a 20 °C incubator with 60% humidity in 12/12 light/dark cycle. For silencing, males that were trained with a single mated female at 20 °C (experienced group) or were left alone (naïve group) were transferred to a 30 °C incubator, or as indicated, with 60% humidity for the specific time period, plus an extra 30 min at the start and the end of the silencing time. Afterward, they were transferred back to the 20 °C incubator. As a temperature control, similar sized groups of experienced and naïve flies were kept in 20 °C during that time.

For silencing during sleep assays, videos of the flies in sleep plates were recorded from both 20 °C and 30 °C groups simultaneously for off-line sleep analysis with SleepTracker. For courtship assays, flies were kept at 20 °C for 24 h until the test.

For monitoring neural activity while silencing the specific neurons in freely behaving animals, males were raised, collected and trained at 20 °C. After training, they were transferred into a luminescence plate and loaded in the Spark microplate reader (Tecan) equipped with a heating but not cooling module. For the experiments, the microplate reader was housed in a 17 °C room to raise the temperature to either 20 °C or 32 °C.

**Calcium imaging**. Calcium imaging was performed on a customized resonant scanning two-photon microscope with near-infrared excitation (920 nm, Spectra-Physics, INSIGHT DS DUAL) and a 25x water immerse objective (Nikon MRD77225 25XW). The microscope was controlled by ScanImage 2016bR0 (Vidrio Technologies). A piezo objective scanner (P-725K129, Physik Instrumente, Germany) with a controller (E-709, Physik Instrumente, Germany) was equipped to enable a whole-brain volume scanning up to 4 hz in flies.

For calcium imaging in explant brains, flies were immobilized on ice and brains were dissected out in extra-cellular saline[47] (103 mM NaCl, 3 mM KCl, 2 mM CaCl₂, 4 mM MgCl₂, 26 mM NaHCO₃, 1 mM NaH₂PO₄, 8 mM trehalose, 10 mM glucose, 5 mM TES, bubbled with 95% O₂/5% CO₂) and mounted anterior side up on a cover slip in a Sylgard-lined dish in a 20 °C saline bath.

For calcium imaging in vivo brain preparations, flies were immobilized on ice and then put in a fly body shape-and-sized hole on the plastic film in a customized chamber[48]. Ultraviolet (UV) curing adhesive (Loctite 352, Henkel) was applied at gaps between the fly and the hole and fixed with a brief (3–5 s) UV irradiation (LED-200 Electrolite). The fly's position was carefully adjusted before the fixation so that the upper half of its head capsule and thorax was above the film and its legs and abdomen could move freely below the film. The dorsal section of the head was bathed in extra-cellular saline, then a rectangular cuticle between the eyes was removed to expose the brain. The air sacs and fat tissues were removed with tweezers. To ensure that the brain remained stationary during imaging, the no. 16 muscles were cut to disable the frontal pulsate organ. The esophagus was also cut and carefully removed via the neck.

To test for functional connectivity between neurons, Chrimson88 and GCaMP6s were expressed in the target upstream and downstream neurons, respectively. Photostimulation light was delivered in a pulse train that consisted of six 5 s pulses (100% duty cycle during each pulse) with a 10 s latency to the first pulse and 30 s interval between pulse onsets, which makes a 200 s session with the light intensity ~ 0.6 mW/mm², as measured using Thorlabs S170C power sensor, which is delivered from a 660 nm LED (M660L3 Thorlabs) coupled to a digital micromirror device (DMD) (Texas Instruments DLPC300 Light Crafter) and combined with the imaging light path using a FF757-DiO1 dichroic (Semrock). On the emission side, the primary dichroic was Di02-R635 (Semrock), the detection arm dichroic was 565DCXR (Chroma), and the emission filters were FF03-525/50 and FF01-625/90 (Semrock). For the GCaMP imaging of the targeted downstream neuron, fly brains were sampled by volumes of 42 frames with 3 μm per step, at approximately 1hz volume rate images with ~157 μm x 157 μm field of view at 512 × 512 pixels resolution for explant brains and with ~157 μm x 78.5 μm field of view at 512 × 256 pixels resolution for in vivo brains. Time series of volume images were acquired for 200 s to cover the whole stimulation session with the excitation laser power of ~12 mW.

For spontaneous activity measurements, jGCaMP7b[49] was expressed in target neurons. Single males from trained or naïve groups were prepared for in vivo imaging in customized imaging chambers and rested for ~30 min before the imaging. Whole brain volumes were sampled 60 s with a ~157 μm x 78.5 μm field of view at 512 × 256 pixels resolution by 15 frames per volume with 5 μm step size at 4 hz with an excitation laser power of ~12 mW. Imaging in experienced and naïve males was performed in random order to minimize temporal variation in spontaneous activity between tested neurons in different flies.

All image data were analyzed off-line with Image J (1.53c, Fiji) and customized MATLAB codes (Github). Region of interest (ROI) of brain regions were manually defined and the average GCaMP signal within the ROI were calculated to represent the activity of targeted neuron. Time series calcium activity was extracted from the image data. To measure functional connectivity, the Calcium activity ΔF/F is defined as ΔF/F = (f(t) − F0)/F0, where the f(t) is the calcium signal intensity and the F0 is the mean F of the first 10 s of the imaging sessions before optogenetic activation. ΔF/F traces of six stimulation were aligned to the LED onset and averaged to represent the targeting neural activity upon neuron activations. To determine a positive connectivity, the mean ΔF/F during 10 s pre-stimulation were taken as baseline activity and the mean ΔF/F during stimulation as stimulated activity in a fly. Groups of baseline activities and stimulated activities of different flies were tested with a Student T test or Wilcoxon Rank Sum test to determine if optogenetic activation had evoked significant calcium activity changes in the targeting neuron against the hypothesis that the baseline activity and stimulated activity were the same level. A P value smaller than 0.05 was taken as the criteria of connectivity (either inhibitory or excitatory).

Spontaneous activity of a neuron is the mean value of calcium intensities during a 60 s recording session.

**Multicolor Stochastic Labelling (MCFO)**. To reveal morphology of the SFSs, MCFO was performed by the Janelia Flylight Team. Detailed protocol can be found online at https://www.janelia.org/project-team/flylight/protocols. The SFS neuron expression pattern was extracted in VVD viewer (Ver. 161214, https://github.com/takashi310/VVD_Viewer).

**EM connectivity**. Neuronal skeletons and connectivity between neurons of interest were extracted from the Focused Ion Beam Scanning Electron Microscopy (FIB-SEM) volume of the *Drosopila* hemibrain (v1.1)[29,30], using neuPrint[50] (https://neuprint.janelia.org/). Synapse numbers between neurons were pulled out with a custom cypher query and further analyzed in Excel (Microsoft Office 365). To determine the connectivity between two cell types, a threshold of 9 synapses was applied to single-neuron pairs and then connections of neurons of the same cell type were grouped together.

**Identification of the GAL4 lines targeting specific neurons**. Skeletons of neurons of interest were extracted from the Janelia hemibrain dataset v1.1 and rendered for demonstration in VVDviewer (Ver. 161214, Otsuna H.). Color depth MIP mask[51] of the neuron were generated from the skeleton and used to search through the Janelia FlyLight Generation1 GAL4/LexA collection[32,52] and FlyLight Generation1 MCFO collection[53,54]. The expression pattern images of selected lines can be viewed at https://flweb.janelia.org/cgi-bin/flew.cgi and https://gen1mcfo.janelia.org/cgi-bin/gen1mcfo.cgi.

**Statistical analysis and data reproducibility**. For statistical analysis of the courtship conditioning data, a MATLAB script (permutation test, github)[55] was used. Briefly, the entire set of courtship indices for both naïve and trained flies were pooled and then randomly assorted into simulated naïve and trained groups of the

same size as the original data. A SI was calculated for each of 100,000 randomly permutated data sets, and $P$ values were estimated for the null hypothesis that learning equals 0 ($H_0$: SI = 0) or for the null hypothesis that experimental and control males learn equally well ($H_0$: $SI_{exp} = SI_{ctrl}$).

A two-sided Student $T$ test was used to compare normalized luminescence data, and GCaMP data between experimental and control groups or between experimental groups sharing the same control group after the dataset was tested and confirmed to be normally distributed by Jarque-Bera test in MATLAB. A two-sided Wilcoxon Signed Rank test was used to test if ΔSleep data, ΔP doze data, ΔP wake data of single group are equal to zero. A two-sided Wilcoxon Rank Sum test was used to compare ΔSleep data, ΔP doze data, ΔP wake data between different groups.

All statistical analyses were done with MATLAB. All behavioral assays were repeated at least 3 times with similar results. All imaging data were collected on independent fly samples.

**Reporting summary**. Further information on research design is available in the Nature Research Reporting Summary linked to this article.

## Data availability
Source data are provided with this paper.

## Code availability
The code used for data collection and analysis is available on the following GitHub page: https://github.com/Kelemanlab

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

## Acknowledgements
The work was supported by Howard Hughes Medical Institute at Janelia Research Campus. We thank Ulrike Heberlein and Bruno van Swinderen for comments on the manuscript, Y. Aso and G. Rubin for SS01306 split line, Tansy Yang (Janelia Connectome Annotation Team) for help with EM connectome search, Otsuna Hideo (Janelia Scientific Computing team) for help with search for specific GAL4 lines, Dan Bushey (Janelia Project Technical Resources) for help with optogenetic activation. This work was made possible in part by software funded by the NIH: Fluorender (VVD viewer): An Imaging Tool for Visualization and Analysis of Confocal Data as Applied to Zebrafish Research, R01-GM098151-01.

## Author contributions
K.K., Z.L. conceived the project and designed the experiments. Z.L. performed most of the experiments, wrote all the Matlab programs and analyzed the data. KH performed and analyzed the data in Figs. 1e, 2c, 3h and 4f. K.K. supervised the project and wrote the manuscript with help of Z.L. and K.H.

## Competing interests
The authors declare no competing interests.
