## [Peer Review File · Nature Communications]

A neural circuit linking learning and sleep in *Drosophila* long-term memoryREVIEWER COMMENTS

Reviewer #1 (Remarks to the Author):

In this study, the authors investigate the mechanisms whereby learning modulates sleep. The authors focus on the interaction between two structures (the Mushroom Body and the ventral Fan Shaped Body) that have been separately linked to sleep or memory. Interestingly, the vFB are only activated by learning experiences that induce long term memory; silencing the vFBs prevented post-training increases in sleep. The authors identify the MBONs- $\gamma 2\alpha'1$ neurons as mediating the increased sleep after LTM. In contrast, MBONs- $\beta'2mp$ were only activated by a short experience and inhibit vFB neurons.

This is a very well written manuscript that addresses an extremely important topic not only for *Drosophila* neurobiology specifically but for neuroscience in general. Indeed, there is an increasing interest in better understanding how the mechanisms underlying sleep and plasticity are intertwined. This work serves as a blueprint that can be used by other investigators to fully characterize how these complex phenomena interact. The experiments are well controlled and the interpretation of the data is measured and consistent with the underlying data. I do not have any major issues.

Reviewer #2 (Remarks to the Author):

In this study, Lei and Keleman sought to understand how courtship learning regulated by the Mushroom Bodies is conveyed to the ventral Fan-Shaped Body (vFB) to induce post-learning sleep. This work follows up a 2019 publication from the same group demonstrating that the vFB both induces sleep and reactivates the memory-inducing dopamine neurons in courtship memory training. The current study elegantly demonstrates that two different type of Mushroom Bodies Output Neurons (MBONS) are involved in conveying learning experience to the vFB. Importantly, the authors also show that both types of MBONS (MBONs- $\gamma 2\alpha'1$ and MBONs- $\beta'2mp$) regulate the activity of the vFB via the SFS interneurons. The authors demonstrate that SFS interneurons regulate the activity of the vFB by integrating excitatory inputs from MBONs- $\gamma 2\alpha'1$ and inhibitory inputs from MBONs- $\beta'2mp$. This is an extremely well-designed and convincing study that will be of great interest to the fly community.

I only have a couple of points to raise:

- In the experiments done using MBONs- $\beta'2mp$ (Figure 3), the authors show that the activity of these neurons is maximal following a 2 hour training session in the time window 7-9h after beginning of training (Figures 3c and d). However, in Figures 3e and 3f, the authors demonstrate that silencing MBONs- $\beta'2mp$ neurons in the time period 2.5-6h following the beginning of a 2-hour training session leads to the formation of LTM. Why the discrepancy in timing of silencing and timing of maximal activity following a 2-hour training session? Is the activity of MBONs- $\beta'2mp$ neurons increased in the time period 2.5-6h following the beginning of a 2-hour training session? The authors should clarify this point.
- Line 61: "We exposed naïve males to mated females for 1, 2, 4 or 6 hours in single pair assays and measured the amount of sleep for the rest of the day as the sleep deprivation during the night does not significantly impair LTM." This sentence is confusing, what are the authors trying to say when mentioning sleep deprivation?
- Maybe the authors could have cited the 2016 Montague and Baker paper "Memory Elicited by Courtship Conditioning Requires Mushroom Body Neuronal Subsets Similar to Those Utilized in Appetitive Memory" as this study demonstrated an involvement for both MBONs- $\gamma 2\alpha'1$ and MBONs- $\beta'2mp$ in courtship STM.

Reviewer #3 (Remarks to the Author):

Sleep is broadly considered to be relevant for memory consolidation. In this manuscript, the authors identify the neural circuit that regulates learning-induced post-training sleep. The study utilized *Drosophila* courtship conditioning to assess the effect of differential training durations on the activity of sleep-promoting vFB neurons. The results indicate that vFB neurons only promote sleep when flies are trained for an extended duration, which results in long-term memory formation. The prolonged learning experience activates MBON- γ 2 α '1 and inhibits MBON- β '2mp neurons. These opposing inputs are integrated by SFS interneurons, which provide input to vFB neurons. Together, this circuit mediates post-training sleep, which is essential for the reactivation of DAN-aSP13 neurons to promote long-term memory formation.

Altogether, the work reveals the circuit basis of the connection between memory and sleep centers in the *Drosophila* brain that gates long-term memory consolidation, which should be of general interest. However, the data on how sleep-deprivation affects the circuit dynamics in trained flies and if SFS interneurons' manipulation affects sleep and memory are not shown, which is essential for the overall conclusions. These and additional concerns are detailed below which should be clarified. Assuming these concerns are addressed satisfactorily, the current research will add significantly to our understanding of learning-induced sleep and its role in memory formation.

Comments

1. Previous work (from Sehgal and Shaw labs) has demonstrated that trained flies show an increase in sleep in the first 4-6h post-training. This also can be observed in Fig.1c in which flies trained for 4h and 6h sleep better than controls in the 7-11h period. However, an increase in vFBs activity can only be observed in the 7-8h period. How vFBs, which are active for an hour post-training, able to promote sleep in the 7-11h period? Also, why was the 7-9h period considered for analysis?
2. One of the main findings is that selective activation of vFBs neurons depends on the intensity of the training experience and therefore, 6h training results in the activation of vFB but not 1h training. Another explanation for the observed differences can be the varying duration of experience with a female during training as this can impact sleep. A good control experiment can be training males with virgin females for the same duration as with a mated female. Also, a 3h massed v spaced courtship conditioning protocol can show if vFBs neurons are specifically required for sleep in flies that form long-term memory.
3. Sleep-deprivation affects memory consolidation including courtship suppression. However, the effect of sleep-deprivation on underlying neural circuits relevant for memory formation has not been shown. Experiments assessing the effect of sleep-deprivation on the activity of MBONs, SFSis, and vFBs in trained flies should be included. This will help in understanding how sleep-deprivation affects courtship suppression.
4. The study shows that an increase in MBON- γ 2 α '1 activity and inhibition of MBON- β '2mp neurotransmission is correlated with vFBs activation in trained flies. To assess if these connections are relevant after conditioning the authors should test the effect of silencing MBON- γ 2 α '1 and MBON- β '2mp post-training on vFBs neurons in trained flies?
5. Can the authors explain why a linear increase in the activity of sleep-promoting MBON- γ 2 α '1 neurons with the duration of training is not resulting in a concomitant increase in post-training sleep?
6. In Fig.3f, the authors show that inhibiting MBON- β '2mp neurotransmission in flies trained for only 2h is sufficient to induce LTM formation. However, 2h training is not sufficient to activate MBON- γ 2 α '1 neurons as shown in Fig.2g. How are flies then able to form LTM considering that both an increase in MBON- γ 2 α '1 and a decrease in MBON- β '2mp neurotransmission are essential for memory consolidation according to the proposed model.
7. Authors should include memory and sleep post-training data with SFSis manipulation as it is essential to determine the relevance of the circuit and therefore, for the overall conclusions.
8. Can the authors add a discussion on how the activity levels of MBON- γ 2 α '1 and MBON- β '2mp neurons are changing based on the intensity of learning and, how exposure to a mated female during conditioning, which represents training intensity, is resulting in post-training activity changes in MBONs?

Minor issues

1. In line 129, the authors indicate that an all-or-none activation of vFBs is mediating memory formation. How an all-or-none switch can explain the difference in memory and post-training sleep between 4h and 6h of training as shown in Fig.1d and Fig.1e?
2. Line 158: MBON- γ 2 α '1 is mislabelled as MBON- γ 2 α 1.
3. In Fig.2a, the MBON- β '2mp label is missing.
4. In Fig.2c, the neurotransmission was blocked in MBON- γ 2 α '1 neurons in the 7-10h period to assess memory while for sleep assessment these neurons were silenced in the 6.5-9.5h period. Why 3h period was used to silence neurons instead of 2h, and why different time-periods were used in Fig.2c and 2d?
5. In Fig.2d, control flies at both temperatures demonstrate a modest, although significant, increase in post-training sleep compared to other flies. However, as shown in Fig.2c, these flies can form strong memory. Similar behavior can be observed in Fig.3e and f. How a modest increase in sleep post-training supports memory formation?
6. A control for Fig.2d should assess the effect of silencing MBON- γ 2 α '1 in naive flies on sleep. A similar control should also be provided for Fig.3e with MBON- β '2mp manipulation.
7. Line 281: α '1 is missing in the legend.
8. Does MBON- γ 2 α '1 neurons activation post-training in flies trained for only 2h enhance courtship suppression?
9. What is the effect of activating MBON- β '2mp on post-learning sleep? Fig.3g shows that activating MBON- β '2mp affects memory but a similar experiment for post-learning sleep is missing.
10. Fig.4 is mislabelled as Fig.3 in the legends.
11. In extended data Fig.2d, GCaMP6s is mislabelled as 6b.
12. Line 417: What is 'Fig.1b' referring to?
13. Line 552: is mean or median was utilized to measure suppression index.
14. Line 560: Why training chambers and fly food were different for luminescence assays compared to behavioral assays. Is it possible to continuously monitor luminescence in flies kept with a mated female?
15. Line 575: is it 10mm diameter?

Response to the reviewers

We thank the reviewers for their helpful comments. We respond to the specific issues as follows:

Reviewer #1

In this study, the authors investigate the mechanisms whereby learning modulates sleep. The authors focus on the interaction between two structures (the Mushroom Body and the ventral Fan Shaped Body) that have been separately linked to sleep or memory. Interestingly, the vFB are only activated by learning experiences that induce long term memory; silencing the vFBs prevented post-training increases in sleep. The authors identify the MBONs- $\gamma 2\alpha'1$ neurons as mediating the increased sleep after LTM. In contrast, MBONs- $\beta'2mp$ were only activated by a short experience and inhibit vFB neurons.

This is a very well written manuscript that addresses an extremely important topic not only for *Drosophila* neurobiology specifically but for neuroscience in general. Indeed, there is an increasing interest in better understanding how the mechanisms underlying sleep and plasticity are intertwined. This work serves as a blueprint that can be used by other investigators to fully characterize how these complex phenomena interact. The experiments are well controlled and the interpretation of the data is measured and consistent with the underlying data. I do not have any major issues.

We appreciate your comments.

Reviewer #2:

In this study, Lei and Keleman sought to understand how courtship learning regulated by the Mushroom Bodies is conveyed to the ventral Fan-Shaped Body (vFB) to induce post-learning sleep. This work follows up a 2019 publication from the same group demonstrating that the vFB both induces sleep and reactivates the memory-inducing dopamine neurons in courtship memory training. The current study elegantly demonstrates that two different type of Mushroom Bodies Output Neurons (MBONS) are involved in conveying learning experience to the vFB. Importantly, the authors also show that both types of MBONS (MBONs- $\gamma 2\alpha'1$ and MBONs- $\beta'2mp$) regulate the activity of the vFB via the SFS interneurons. The authors demonstrate that SFS interneurons regulate the activity of the vFB by integrating excitatory inputs from MBONs- $\gamma 2\alpha'1$ and inhibitory inputs from MBONs- $\beta'2mp$. This is an extremely well-designed and convincing study that will be of great interest to the fly community.

I only have a couple of points to raise:

- In the experiments done using MBONs- $\beta'2mp$ (Figure 3), the authors show that the activity of these neurons is maximal following a 2 hour training session in the time window 7-9h after beginning of training (Figures 3c and d). However, in Figures 3e and 3f, the authors demonstrate that silencing MBONs- $\beta'2mp$ neurons in the time period 2.5-6h following the beginning of a 2-hour training session leads to the formation of LTM. Why the discrepancy in timing of silencing and timing of maximal activity following a 2-hour training session? Is the activity of MBONs-

$\beta'2mp$ neurons increased in the time period 2.5-6h following the beginning of a 2-hour training session? The authors should clarify this point.

To clarify the timing of the post-learning time periods after each training, we now measure the post-learning time from the end, instead of the start, of each training. Given that trainings of different duration are all aligned to the end of the 6-hour training (from -6 to 0 hours) the post-learning time period after each training starts at 0 hour (line 42-44). Thus, the previously labelled TP7-9 post-training time window, is now depicted as TP1-3 after each training.

In Fig. 3e and d we mistakenly labelled the silencing period as 2.5-6 hours instead of 2.5-5.5 (now TP0.5-3.5 hours). To silence neurons in TP1-3 when their activity after 2-hour training is the highest, we incubated males at 30C during that time. Given the rather low resolution of shi^{ts} in time and to ensure we fully cover that time period, we extended the 30C incubation for 30 minutes at each end, hence the TP0.5-3.5 (previously 2.5-5.5 hours). Thus, there is really no discrepancy in timing of silencing and timing of the MBONs- $\beta'2mp$ maximal activity following a 2-hour training session.

We now provide the information, as to why we use the 3-hour silencing period, in the text (line 60-65).

- Line 61: “We exposed naïve males to mated females for 1, 2, 4 or 6 hours in single pair assays and measured the amount of sleep for the rest of the day as the sleep deprivation during the night does not significantly impair LTM.” This sentence is confusing, what are the authors trying to say when mentioning sleep deprivation?

In our previous work (Dag et al., *Elife* 2019) we showed that sleep deprivation in the specific post-training time window during the day but not during the night impairs courtship long-term memory. These data suggested that the night sleep has no role in memory consolidation. Therefore, to assess the amount of sleep after varying amount of training we focused only on sleep during the day.

We agree that this sentence is confusing. To clarify it, we removed the reference to the night sleep in the text as it is not relevant in that context.

- Maybe the authors could have cited the 2016 Montague and Baker paper “Memory Elicited by Courtship Conditioning Requires Mushroom Body Neuronal Subsets Similar to Those Utilized in Appetitive Memory” as this study demonstrated an involvement for both MBONs- $\gamma2\alpha'1$ and MBONs- $\beta'2mp$ in courtship STM.

Thank you for pointing this out. We now cited the 2016 Montague and Baker paper (line 215).

Reviewer #3:

Sleep is broadly considered to be relevant for memory consolidation. In this manuscript, the authors identify the neural circuit that regulates learning-induced post-training sleep. The study utilized *Drosophila* courtship conditioning to assess the effect of differential training durations on the activity of sleep-promoting vFB neurons. The results indicate that vFB neurons only promote sleep when flies are trained for an extended duration, which results in long-term memory formation. The prolonged learning experience activates MBON- $\gamma2\alpha'1$ and inhibits MBON- $\beta'2mp$ neurons. These opposing inputs are integrated by SFS interneurons, which provide input to vFB neurons. Together, this circuit mediates post-training sleep, which is

essential for the reactivation of DAN-aSP13 neurons to promote long-term memory formation. Altogether, the work reveals the circuit basis of the connection between memory and sleep centers in the *Drosophila* brain that gates long-term memory consolidation, which should be of general interest. However, the data on how sleep-deprivation affects the circuit dynamics in trained flies and if SFS interneurons' manipulation affects sleep and memory are not shown, which is essential for the overall conclusions. These and additional concerns are detailed below which should be clarified. Assuming these concerns are addressed satisfactorily, the current research will add significantly to our understanding of learning-induced sleep and its role in memory formation.

Comments

1. Previous work (from Sehgal and Shaw labs) has demonstrated that trained flies show an increase in sleep in the first 4-6h post-training. This also can be observed in Fig.1c in which flies trained for 4h and 6h sleep better than controls in the 7-11h period. However, an increase in vFBs activity can only be observed in the 7-8h period. How vFBs, which are active for an hour post-training, able to promote sleep in the 7-11h period? Also, why was the 7-9h period considered for analysis?

To clarify the timing of the post-learning time periods after each training, we now measure the post-learning time from the end, instead of the start, of each training. Given that trainings of different duration are all aligned to the end of the 6-hour training (from -6 to 0 hours) the post-learning time period after each training starts at 0 hour (line 42-44). Thus, the previously labelled TP7-9 post-training time window, is now depicted as TP1-3 after each training.

We have previously established that activity of vFBs is essential for the long-term memory consolidation in TP1-3 (previously TP7-9), but not during other time intervals post-training (Dag et al., *Elife* 2019). We also determined that dopaminergic neurons, DAN- γ 5, which are essential for courtship memory acquisition are reactivated by vFBs in TP1-3 for long-term memory consolidation (Dag et al., *Elife* 2019). Therefore, in this study we focused our analysis on that specific time window after learning. We note this in the text (line 25-27 and 32-35).

Increase in the vFBs activity (measured with the luminescence assay) after 6- and 4-hour training is significantly higher in TP1-3 relative to naïve males and those trained for 1 or 2 hours. (entirely in TP1-2 and partially in TP2-3). In contrast, significant enhancement of the post-learning sleep is the highest in both TP1-2 and TP2-3. That difference is very likely a result of the lower sensitivity of the luminescence-based method we have used to measure activity of vFBs in freely behaving animals. Indeed, imaging of Ca^{++} levels with the fluorescent GCaMP7b in *in vivo* brain preparations, has revealed that activity of vFBs is significantly higher after 6-hour training in the entire TP1-3 relative to naïve males and males that were trained for 1 hour (Extended Data Fig. 2b). We now make a note of this in the text (line 52-57).

Based on these results we conclude that vFBs exhibit significantly higher activity level in the entire TP1-3 period and hence sustains post-learning sleep during this time. That post-learning sleep seems to persist beyond 2 hours which might be a result of prolonged vFB activity, but beyond detection by our assays.

2. One of the main findings is that selective activation of vFBs neurons depends on the intensity of the training experience and therefore, 6h training results in the activation of vFB but not 1h training. Another explanation for the observed differences can be the varying duration of experience with a female during training as this can impact sleep. A good control experiment can be training males with virgin females for the same duration as with a mated female. Also, a 3h massed v spaced courtship conditioning protocol can show if vFBs neurons are specifically required for sleep in flies that form long-term memory.

In our previous work (Dag et al., *Elife* 2019) we have addressed that issue and discussed in the first version of the manuscript (line 27-30). Briefly, to test whether the enhanced post-learning sleep is a result of the extended courtship learning experience or just extended courtship behavior, we trained either wild type or mutants males for dopamine receptor, *DoR1*, to induce long-term memory. Wild type males, which learn and form long-term memory, displayed an enhanced post-learning sleep. However, *DopR1* mutant males which can't learn (Keleman et al., *Nature* 2012) although, they courted mated females as much or even more than the wild type males they neither exhibited enhanced post-learning sleep nor long-term memory. All together, these results strongly suggested that the extended learning, not just extended courtship, induces post-learning sleep.

Using receptive virgin female as trainers to dissociate learning from the courtship behavior itself is a great idea. But since virgin females are receptive, it is not possible to use them in a prolonged training paradigm. They become unreceptive shortly after copulation and induce learning and hence the memory. Similarly, to distinguish between long-term and other forms of memory using mass training, it is not possible in the context of courtship learning. During courtship learning, males court unreceptive females always in an ON and OFF fashion, which closely resemble a spaced training.

Since post-learning sleep is induced after learning experience that leads to LTM but not STM (Fig. 1 c and d) and vFBs are essential for post-learning sleep after training to induce LTM (Fig. 1f) we concluded that vFBs are specifically required for sleep in flies that form LTM.

However, to strengthen our conclusions, we tested the requirement of vFBs for both types of memory. We trained males either for 1 or 6 hours with mated females to induce either short- or long-term memory. Silencing of vFBs impaired formation of long but not short-term memory. We present these data in the Ext. Data Fig. 2g, 2h, line 60-67)

3. Sleep-deprivation affects memory consolidation including courtship suppression. However, the effect of sleep-deprivation on underlying neural circuits relevant for memory formation has not been shown. Experiments assessing the effect of sleep-deprivation on the activity of MBONs, SFSs, and vFBs in trained flies should be included. This will help in understanding how sleep-deprivation affects courtship suppression.

Previously, we established that deprivation of the post-learning sleep in TP1-3, but not other time windows, impairs LTM formation (Dag et al., *Elife* 2019). We assume that the reviewer would like to see the effect the sleep deprivation in TP1-3 might have on the activity of the specific neurons within the circuit during this time. Although we and others at Janelia Research Campus work very hard to develop the luminescence assay further, at present, we have no means to deprive males of sleep while monitoring activity of the specific neurons in that circuit. But since post-learning sleep is a consequence not a cause of that circuit activity, such an experiment might be not very informative.

4. The study shows that an increase in MBON- $\gamma 2\alpha'1$ activity and inhibition of MBON- $\beta'2mp$ neurotransmission is correlated with vFBs activation in trained flies. To assess if these connections are relevant after conditioning the authors should test the effect of silencing MBON- $\gamma 2\alpha'1$ and MBON- $\beta'2mp$ post-training on vFBs neurons in trained flies?

To test whether increased MBON- $\gamma 2\alpha'1$ activity after learning experience to induce LTM leads to enhanced vFBs activity, we identified a specific MBON- $\gamma 2\alpha'1$ -LexA line to silence MBON- $\gamma 2\alpha'1$ while simultaneously monitoring activity in vFBs with the luminescence reporter expressed with GAL4/UAS system. Consistent with our model, silencing of the excitatory MBON- $\gamma 2\alpha'1$ input to vFBs, blocked the vFBs activity increase in males trained for 6 hours, which leads to LTM, relative to males with MBON- $\gamma 2\alpha'1$ functional and the genetic controls. (Ext. Fig. 1j and 3a, line 127-130).

We were unable to test directly whether activity of MBONs- $\beta'2mp$ is essential for vFBs inhibition after training for STM. We could not silence those neurons exclusively while simultaneously monitoring activity of vFBs, as the only available MBONs- $\beta'2mp$ -LexA line was also expressed in MBONs- $\gamma 5\beta'2a$ which are essential for memory acquisition (X. Zhao et al., *Elife*, 2018) (Ext. Data Fig. 1k). However, we believe that our new (Fig. 3h) and previous experiments (Fig. 3e, 3f, 3g, 3i) that showed that MBONs- $\beta'2mp$ silencing after training for STM induced post-learning sleep and LTM while activation after training for LTM impaired both sleep and memory strongly suggest that the inhibitory connection between MBONs- $\beta'2mp$ and vFBs inhibits sleep after learning for STM and hence prevents such an experience to persist (line 161-176).

5. Can the authors explain why a linear increase in the activity of sleep-promoting MBON- $\gamma 2\alpha'1$ neurons with the duration of training is not resulting in a concomitant increase in post-training sleep?

Based on our results we postulate a following model of how the MB/vFB neural circuit controls sleep induced by learning. The sleep promoting excitatory MBON- $\gamma 2\alpha'1$ neurons become increasingly active post-learning with the increasing time of training. However, post-learning sleep is only enhanced after prolonged training, 4 or 6 hours. This is due to the inhibitory input of the sleep suppressing MBON- $\beta'2mp$ neurons but only after short-term training, eg., after 1 or 2 hours. SFS neurons which integrate both MBON- $\gamma 2\alpha'1$ excitatory and MBON- $\beta'2mp$ inhibitory inputs convey it onto vFBs. As a result, the SFS neurons and hence vFBs are activated only after the inhibition has been diminished and the excitation level reached a certain level, which is after 4 or 6 hours of training. We illustrate this model in Fig. 4j (line 207-222).

6. In Fig.3f, the authors show that inhibiting MBON- $\beta'2mp$ neurotransmission in flies trained for only 2h is sufficient to induce LTM formation. However, 2h training is not sufficient to activate MBON- $\gamma 2\alpha'1$ neurons as shown in Fig.2g. How are flies then able to form LTM considering that both an increase in MBON- $\gamma 2\alpha'1$ and a decrease in MBON- $\beta'2mp$ neurotransmission are essential for memory consolidation according to the proposed model.

The sleep promoting MBON- $\gamma 2\alpha'1$ neurons are activated with the increasing time of training in a linear fashion. They are slightly but significantly more active after 2-hour training relative to males that were not trained (Fig. 2 f and g). However, they are unable to induce sleep because of

the activity of the MBON- β '2mp neurons, which inhibit excitatory input of the MBON- γ 2 α '1 neurons, after training for 2 hours, in SFS neurons and hence sleep and memory consolidation. Removing that inhibition by silencing of the inhibitory MBON- β '2mp neurons after training for 2 hours enables the excitatory MBON- γ 2 α '1 to activate SFS and hence vFBs. In turn, that leads to the post-learning sleep enhancement and consolidation of the 2-hour learning experience into long-term memory.

7. Authors should include memory and sleep post-training data with SFSis manipulation as it is essential to determine the relevance of the circuit and therefore, for the overall conclusions. These are very important points. As suggested, we silenced SFS neurons after 6-hour training and monitored sleep and LTM. Consistent with our model, silencing of SFS neurons after prolonged training, prevented post-learning sleep enhancement and LTM formation. These results suggest that SFS neurons are essential for the post-learning sleep enhancement to consolidate LTM after prolonged learning (Fig. 4f, 4g, line 193-197). Moreover, these results support our model that SFS neurons are the intermediate neurons that integrate two opposing MB outputs and convey it on vFBs to induce post-learning sleep after prolonged learning experience.

8. Can the authors add a discussion on how the activity levels of MBON- γ 2 α '1 and MBON- β '2mp neurons are changing based on the intensity of learning and, how exposure to a mated female during conditioning, which represents training intensity, is resulting in post-training activity changes in MBONs?

We have expanded our discussion of how activity of MBONs changes upon learning experience with mated females (line 224-235)

Minor issues

1. In line 129, the authors indicate that an all-or-none activation of vFBs is mediating memory formation. How an all-or-none switch can explain the difference in memory and post-training sleep between 4h and 6h of training as shown in Fig.1d and Fig.1e?

We refer to an all-or-none activation of vFBs in the step-wise fashion after prolonged learning experience. We now clarified that point (line 142-143)

2. Line 158: MBON-- γ 2 α '1 is mislabelled as MBON-- γ 2 α 1.

We have corrected our mistake.

3. In Fig.2a, the MBON- β '2mp label is missing.

To identify the MBON- β '2mp neurons in Fig. 2a, we have added the ID number of the MBON- β '2mp neurons in the text (line 147).

4. In Fig.2c, the neurotransmission was blocked in MBON- γ 2 α '1 neurons in the 7-10h period to assess memory while for sleep assessment these neurons were silenced in the 6.5-9.5h period. Why 3h period was used to silence neurons instead of 2h, and why different time-periods were used in Fig.2c and 2d?

We mistakenly labelled in Fig. 2c the silencing period as TP7-10 (now TP1-4), instead of TP6.5-9.5 (now TP05.3.5). To silence the MBONs- γ 2 α '1 neurons in TP1-3, we incubated males at 30C

during that time. Given the low resolution of shifts in time, and to assure we fully cover that time-period, we extended the 30C incubation for 30 minutes at each end, hence the TP0.5-3.5 (previously 6.5-9.5 hours). So, there is really no discrepancy in silencing of the MBONs- $\gamma 2\alpha' 1$ neurons in Fig. 2 c and d.

We now corrected the schematic of the experimental design In Fig. 2c and 2d and we provide the information why we use the 3-hour silencing period in the text (line 60-65) and methods section (line 576-581).

5. In Fig.2d, control flies at both temperatures demonstrate a modest, although significant, increase in post-training sleep compared to other flies. However, as shown in Fig.2c these flies can form strong memory. Similar behavior can be observed in Fig.3e and f. How a modest increase in sleep post-training supports memory formation?

Suppression index (SI), which represents memory, reflects an increase in courtship suppression between groups of trained and naïve males. Likewise, post-learning sleep enhancement, represented by Δ Sleep, reflects an increase in amount of post-learning sleep between groups of trained and naïve males. Therefore, it is not possible to quantitatively compare the amount of post-learning sleep and the strength of memory which was assessed between different experimental groups.

6. A control for Fig.2d should assess the effect of silencing MBON- $\gamma 2\alpha' 1$ in naïve flies on sleep. A similar control should also be provided for Fig.3e with MBON- $\beta' 2mp$ manipulation.

We of course included naïve controls in our experiments. Δ Sleep in Fig. 2d and Fig. 3e reflects the difference in sleep between trained and naïve groups upon neuronal silencing.

7. Line 281: a' 1 is missing in the legend.

We have corrected our mistake.

8. Does MBON- $\gamma 2\alpha' 1$ neurons activation post-training in flies trained for only 2h enhance courtship suppression?

Consistent with our earlier data that post-learning sleep is sufficient to consolidate STM to LTM (D. Ugur et al., ELife, 2019), activation of sleep promoting MBONs- $\gamma 2\alpha' 1$ with csChrimson after 2-hour training, which normally leads to STM only, induced LTM, expressed as a significant courtship suppression 24 hours later relative to controls. We have decided not to include these data in the manuscript as the sufficiency of post-learning sleep in consolidation of STM to LTM would add another level of complexity which is beyond the scope of this study.

Activation of MBONs-g2a'1 is sufficient to consolidate STM into LTM

LTM, represented as SI [%], of males of indicated genotypes after being trained in single-pair assays with mated Canton-S females and MBONs-g2a'1 activated with csChrimson as indicated. P value is for H0 SI=0, ns >0.05, **P < 0.01 and for H0 SI_{exp}=SI_{ctrl}; ns P > 0.05, *P < 0.05, Permutation test.

Genotype:

MBON-γ2α'1-GAL4>UAS-csChrimson	UAS-csChrimson/Y; MB077B-GAL4p65adz/+; MB077B-ZpGAL4dbd/+
pBDP-GAL4>UAS-csChrimson	UAS-csChrimson/Y; +/+; pBDP-GAL4/+

Data table

CHR light	Genotype	CI [%]	n	SI [%]	P SI=0	P SI _L =SI _{L+}	Test
-	MBON-γ2α'1-GAL4>UAS-csChrimson	23.97	51				
		25.60	47	-6.77	0.5130		Permutation test
+	MBON-γ2α'1-GAL4>UAS-csChrimson	31.03	60				
		20.73	57	33.18	0.0013	0.0106	Permutation test
-	pBDP-GAL4> UAS-csChrimson	28.73	53				
		24.51	49	14.67	0.1284		Permutation test
+	pBDP-GAL4> UAS-csChrimson	23.19	50				
		21.49	45	7.33	0.3197	0.7081	Permutation test

9. What is the effect of activating MBON-β'2mp on post-learning sleep? Fig.3g shows that activating MBON-β'2mp affects memory but a similar experiment for post-learning sleep is missing.

Optogenetic activation of MBONs-β'2mp and thus inhibiting vFBs in TP1-3 after training for 6 hours, suppressed sleep (Fig. 3h, line 170-172)

10. Fig.4 is mislabelled as Fig.3 in the legends.

We have corrected our mistake.

11. In extended data Fig.2d, GCaMP6s is mislabelled as 6b.

We have corrected our mistake.

12. Line 417: What is 'Fig.1b' referring to?

It refers to Fig. 1e.

13. Line 552: is mean or median was utilized to measure suppression index.

All SIs are derived from a median as described in the methods section (line 511-517) and shown in the Supplementary table 3.

14. Line 560: Why training chambers and fly food were different for luminescence assays compared to behavioral assays. Is it possible to continuously monitor luminescence in flies kept with a mated female?

The chambers to monitor luminescence differ from the chambers used for training because of different technical specifications needed to measure the luminescence signal. It is not possible to measure luminescence signal in the presence of trainer females as they frequently move and often block the luminescence signal emitted by males.

15. Line 575: is it 10mm diameter? Yes, we have corrected our mistake (line 538).

REVIEWER COMMENTS

Reviewer #2 (Remarks to the Author):

The authors answered all my concerns.

Reviewer #3 (Remarks to the Author):

The authors have addressed my concerns with new experiments and additions to the manuscript. In particular, Extended data Fig. 3 shows that manipulating MBON- $\gamma 2\alpha'1$ activity in trained flies affects vFB activity indicating a training-dependent connection between MBON and vFB neurons. Authors also demonstrate that vFBs are specifically required for LTM but not STM in Extended data Fig. 2. Further, experiments showing the role of SFS neurons in post-training sleep and memory in Fig. 4 support their overall conclusions. In addition, changes in the schematics depicting post-learning time intervals are helpful. This work adds significantly to our understanding of sleep and memory coupling and I fully support this publication.